# Genome-Wide Analysis and Expression Profiling of the Phospholipase D Gene Family in *Solanum tuberosum*

**DOI:** 10.3390/biology10080741

**Published:** 2021-08-02

**Authors:** Long Li, Chao Zhang, Mancang Zhang, Chenghui Yang, Yanru Bao, Dongdong Wang, Qin Chen, Yue Chen

**Affiliations:** 1State Key Laboratory of Crop Stress Biology for Arid Areas, College of Agronomy, Northwest A&F University, Yangling 712100, China; 2019055061@nwafu.edu.cn (L.L.); zhangchao520@nwafu.edu.cn (C.Z.); 2018050059@nwafu.edu.cn (M.Z.); 2018060035@nwafu.edu.cn (C.Y.); baoyanru@nwafu.edu.cn (Y.B.); dongdong-1025@hotmail.com (D.W.); 2College of Food Science and Engineering, Northwest A&F University, Yangling 712100, China

**Keywords:** potato, PLD, abiotic stress, gene expression

## Abstract

**Simple Summary:**

Phospholipase D is the most important kind of phospholipase in plants, which can specifically catalyze the hydrolysis of phosphodiester bonds at the end of phospholipid molecules to produce phospholipid acid and a free group. Phospholipase D is not only involved in maintaining the structural stability of plant cell membrane and the dynamic balance of lipid composition but also involved in a variety of physiological processes, such as plant growth and development and stress response, including stomatal closure, root elongation, cell senescence, high salt stress, cold stress, and drought stress. It plays a critical role in plant growth and development, as well as in hormone and stress responses. In this study, the basic information on phospholipase D gene family was comprehensively introduced, and the physical and chemical properties, systematic evolution, gene structure, conservative motif, chromosome location, gene replication, cis-acting element prediction, GO annotation, interspecific collinearity, and quantitative analysis of potato phospholipase D gene family were analyzed to gain a better understanding of potato phospholipase D gene family. Through the analysis of potato phospholipase D gene family, our results lay the foundation for further understanding of the function of phospholipase gene family in higher plants.

**Abstract:**

Phospholipase D (PLD) is the most important phospholipid hydrolase in plants, which can hydrolyze phospholipids into phosphatidic acid (PA) and choline. When plants encounter low temperature, drought and high salt stress, phospholipase D and its products play an important role in regulating plant growth and development and coping with stress. In this study, 16 members of StPLD gene family were identified in potato genome, which were distributed in α, β, δ, and ζ subfamilies, and their expression patterns under salt, high temperature, drought, and ABA stress were detected by qRT-PCR method. Gene expression analysis showed that the expression of StPLD genes in potato was upregulated and downregulated to varying degrees under the four stresses, indicating that the PLD gene family is involved in the interaction of potato plant hormones and abiotic stress signals. Chromosome distribution showed that StPLD gene was unevenly distributed on 8 chromosomes, and only one pair of tandem repeat genes was found. All StPLD promoters contain hormone and stress-related cis-regulatory elements to respond to different stresses. Structural analysis showed that StPLD genes in the same subgroup had a similar exon–intron structure. Our study provides a valuable reference for further research of the function and structure of PLD gene.

## 1. Introduction

Phospholipase is a class of enzymes responsible for the metabolism and synthesis of phospholipids in plants. They can catalyze the hydrolysis of phospholipids. According to the position where phospholipases hydrolyze phospholipids, they can be divided into five types: phospholipase A1, phospholipase A2, phospholipase B, phospholipase C, and phospholipase D [1]. The most important kind of phospholipase in plants is phospholipase D, which can hydrolyze phospholipids to generate phosphatidic acid and free alcohol groups and is believed to play an important role in many cellular processes. In plants, PLD is involved in some physiological processes, such as cell membrane degradation, signal transduction, vesicle transport, and membrane metabolism during senescence, and abiotic stress damage during seed germination [2,3], including stomatal closure [4], root elongation [5], cold, drought and salt stress [6,7,8]. The function of PLD is usually achieved through the enzymatic hydrolysis product of PA. PA is considered to be a ubiquitous lipid signal molecule, which keeps the cell membrane in a stable state and participates in regulating the dynamic balance of cell membrane lipid composition. Plant PLD is a complex gene family. As early as 1940, PLD was identified as an enzyme involved in lipid metabolism and membrane remodeling [9]. Until 1994, the first PLD encoding cDNA was isolated from the castor bean [10]. At present, the PLD gene family has been found in *Arabidopsis thaliana* [11], *Oryza sativa* [12], *Populus* [13], *Vitis vinifera* [13], *Cicer arietinum* [14], *Camellia sinensis* [15], and other plant species.

In dicotyledons and monocotyledons, the PLD family can be further divided into two subfamilies according to the composition of their N-terminal phospholipid-binding domains, namely, C2-PLD and PXPH-PLD subfamilies. The N-terminus of the C2-PLD subfamily contains the C2 domain, while the N-terminal of the PLD of the PXPH-PLD subfamily contains both Phox homology (Px) domain and Pleckstrin homology (PH) domain [16]. All PLD genes include two HxKxxxxD motifs (motifs), namely HKD1 and HKD2 and C2 conserved domain at N-terminal. HKD is histidine (His), lysine (Lys) and aspartic acid (Asp) [17]. These two motifs are the marker sequences of PLD. The HKD domain endows the enzyme with hydrolytic activity, and PLD in most plants contains a C2 domain in order to better bind to the cofactor Ca^2+^ and enhance the affinity of the enzyme to its substrate [18]. The difference is that in Arabidopsis and rice, it is found that some C2 domains in the PLD gene family are replaced by tandem PX-PH domains [19]. The PX and PH domains of PXPH-PLD have shown a membrane targeting effect, which is closely related to the inositol polyphosphate signal [20]. Protein domain analysis shows that in addition to C2-PLD and PXPH-PLD, there is a new subfamily in rice-SP-PLD. SP-PLD contains signal peptides at the N-terminal. Instead of C2 or PXPH domain [12], its specific cellular localization may be related to plant growth and defense. The activity of PLD will increase under stress, and PA, the product of PLD, also plays an important role in the growth and development of plants [21].

At present, the PLD gene has been found in many plants, 12 PLD genes have been identified in *Arabidopsis thaliana* [22], 17 PLD genes in *Oryza sativa* [12], 18 PLD genes in *Glycine max* [19], and 13 PLD genes in *Zea mays* [2]. According to the sequence characteristics, the PLD gene family can be divided into seven subtypes: PLDα, PLDβ, PLDγ, PLDδ, PLDε, PLDζ, and PLDφ, of which PLDα is the most common. In *Arabidopsis thaliana*, PLDα1 is the most abundant PLD, which has been found to regulate salt tolerance and permeability, abscisic acid signal, and seed aging [23]. PLDα1 is mainly responsible for the activity of PLDs [24]. The study found that in Arabidopsis flowers, pollen tubes, pods, seeds, stems, leaves, roots, and other tissues, except for pollen tubes, AtPLDα1 showed higher expression levels than in other tissues. AtPLDα2, AtPLDβ1, AtPLDβ2, and AtPLDδ were highly expressed in pollen tubes, but low in other tissues. AtPLDγ2 was evenly expressed and not high in these tissues, which indicates the function of different PLD genes. The expression of some PLD genes will change only when they are stimulated by external stimuli [25].

In this study, we identified the potato PLD gene family and obtained 16 potato PLD genes. The physical and chemical properties, gene structure, conservative motif, chromosome location, gene replication, phylogenetic evolution, cis-regulatory element prediction, GO annotation, interspecies collinearity, and qRT-PCR analysis of potato StPLD protein were studied by bioinformatics analysis, which provided some clues for further revealing the function of potato PLD gene family.

## 2. Materials and Methods

### 2.1. Plant Materials Preparation

In this study, the potato variety Desiree was used as experimental material. The experiment was carried out at the State Key Laboratory of Crop Stress Biology in Arid Areas, Northwest A&F University (107°59′–108°08′ east longitude, 34°14′–34°20′ north latitude). The tissue culture seedlings were grown in Murashige and Skoog (MS) medium containing 2% sucrose and 0.8% agar and 0.05% MES (2-Morpholinoethanesulfonic Acid) at pH 5.8 [26]. The study period was from September 2020 to January 2021. After the tissue-cultured seedlings were grown on solid MS medium for one month, selected the same tissue-cultured seedlings to grow on 2% MS nutrient solution medium for one month, and then performed a different treatment. All the tissue culture seedlings were grown in 22 °C, 16 h light (10,000 Lx) and dark 8 h, relative humidity 70% incubator. After growing for one month, the following treatments were carried out, respectively: NaCl treatment, the tissue-cultured seedlings were moved to the nutrient solution containing 200 mM NaCl and soaked for 24 h; high temperature treatment, the tissue-cultured seedlings grown for one month were transferred to 38 °C for 6 h; drought treatment, the tissue-cultured seedlings were moved into a nutrient solution containing 20% polyethylene glycol (PEG6000) and soaked for 24 h; ABA treatment, the tissue-cultured seedlings that had grown for one month were soaked in 2% MS nutrient solution, then 100 μM abscisic acid (ABA) was added and cultured for 3 h. Untreated plantlets were used as controls. The whole treated and control potato plantlets were collected for RNA extraction. For each treatment condition, three biological replicates were established to reduce the error rate and a collection of samples from four potato plantlets were used as one biological replicate [26].

### 2.2. Identification and Phylogenetic Analysis of PLD Gene Family in Potato

Using the protein sequences of PLD gene family identified in *Arabidopsis thaliana* (https://www.arabidopsis.org/index.jsp (accessed on 21 November 2020)) and rice (http://rice.plantbiology.msu.edu/ (accessed on 21 November 2020)) genome database as query sequences, the potato genome database was searched by local Blastp (http://solanaceae.plantbiology.msu.edu/blast.shtml (accessed on 21 November 2020)), and the sequence information of the potato homologous StPLD gene family members was obtained (Table A2) [12]. The domain (PF00614) model file of PLD gene family from Pfam database (http://pfam.xfam.org/ (accessed on 21 November 2020)) was downloaded, HMMER software was used to screen the potato protein sequence containing PLD domain, then the SMART (http://smart.embl-heidelberg.de/ (accessed on 21 November 2020)) and Pfam (http://pfam.xfam.org/search (accessed on 21 November 2020)) websites were used to compare the selected protein sequence [13], and then, in order to test whether the initial identification result was correct, the InterproScan program was used to further screen to confirm the existence of two conservative HKD domains [19]. Prediction and analysis of physicochemical properties of all PLD protein sequences in potato were made using the Expasy website (https://web.expasy.org/protparam/ (accessed on 21 November 2020)). ClustalW was applied to align StPLD, AtPLD, and OsPLD proteins [27] followed by MEGA 10.1.7 to establish the phylogenetic tree with neighbor-joining, bootstrap replicates (1000), p-distance [28].

### 2.3. Chromosomal Location Analysis and Gene Duplication

The chromosome distribution of potato PLD gene was analyzed and mapped by TBtools v1.045 software. The gff3 file of the gene was downloaded from the potato genome database (http://solanaceae.plantbiology.msu.edu/ (accessed on 23 November 2020)) and the gene location information was obtained. Two genes in the same family, the number of other genes was separated by less than or equal to 5, and the distance between the two genes was less than 100 kb, and it was a tandem replicated gene [29].

### 2.4. Gene Structure and Cis-Acting Elements Analysis and Conserved Motif Identification

The MEME website (http://meme-suite.org/ (accessed on 23 November 2020)) was used to analyze the conserved motifs of the potato PLD protein [13]. The storage file of the distribution of exons and introns of the potato PLD gene was downloaded from the potato genome database website, and the TBtools software was used to draw the gene structure map.

In order to study the cis-acting elements in the promoter region of the potato PLD gene, the 1500 bp sequence before the start codon of the PLD gene was retrieved from the potato genome database and submitted to the PlantCARE online website (http://bioinformatics.psb.ugent.be/webtools/plantcare/html/ (accessed on 23 November 2020)) to predict the cis-acting elements [3], and the GSDS2.0 online website (http://gsds.gao-lab.org/index.php (accessed on 23 November 2020)) was used to draw a distribution map of cis-type components.

### 2.5. Gene Ontology Annotations and Interspecific Collinearity Analysis of PLD Gene in Potato

The gene ontology (GO) analysis of potato PLD proteins was performed using the Blast2GO Software (https://www.blast2go.com/ (accessed on 26 November 2020)). The full-length amino acid sequences of 16 StPLD proteins were uploaded to the program with the NCBI database chosen as the reference to analyze three categories: biological process, molecular function, and cellular component [30]. The TBtools software was used to draw the PLD gene collinearity map of potato and Arabidopsis.

### 2.6. Tissue Expression and Stress Treatment Expression Analysis of the Potato PLD Genes

We downloaded the potato PLD family gene fragments per kilobase per million (FPKM) values from the PGSC database. After calculating the value of Log2, the expression heat maps of different tissues and different treatments of PLD gene family were drawn by TBtools [31]. According to the gene expression value downloaded from the PGSC website (http://solanaceae.plantbiology.msu.edu/dm_v6_1_download.shtml (accessed on 26 November 2020)), the tissue-cultured seedlings were treated with 150 mM NaCl, 260 µM Mannitol, 38 °C for 24 h, and 50 µM ABA, 10 µM IAA, 50 µM GA3, and 10 µM BAP for 24 h. The mixed samples were taken after the leaves were infected with P. infestans for 24/48/72 h, the mixed samples were taken after the leaves were treated with BABA for 24/48/72 h, and the mixed samples were taken after the leaves were treated with BTH for 24/48/72 h, the whole plant was used for the expression analysis under the stress conditions.

### 2.7. RNA Isolation and qRT-PCR Analysis

Total RNA from potato was extracted using the RNAsimple Total RNA Extraction Kit (TIANGEN, Beijing, China, DP419) and then Fast Super RT Kit cDNA with gDNase used to perform reverse transcription, according to the manufacturer’s instructions. The design of StPLD gene-specific primers for quantitative real-time PCR (qRT-PCR) analysis was conducted using Primer Premier 5 software, and the ef1α gene was used to normalize the results (Table A1). qRT-PCR was performed on the Q7 Real-Time PCR System. In qRT-PCR experiments, the following thermal cycling conditions were applied: initial activation 95 °C for 30 s, then 40 cycles of 95 °C for 5 s, 60°C for 30 s, and 72 °C for 30 s. The relative expression levels were calculated using the comparative2^−^^ΔΔCT^ method [32].

## 3. Results

### 3.1. Identification of Members of PLD Gene Family in Potato

Using the PLD gene family protein sequences of *Arabidopsis thaliana* and rice, a Blastp search was carried out in the potato genome database and then screened by HMMER software according to the PLD gene domain (PF00614). Then the selected protein sequences were compared by SMART and Pfam, and the genes without the HxKxxxxD domain and only one HxKxxxxD domain were removed. Sixteen PLD genes were identified from the potato genome (Table 1). Finally, in order to verify the correctness of the initial identification results, the InterproScan program was used to further screen to confirm the existence of two conserved HKD domains. The size of the ORF for StPLD proteins varied from 2268 bp to 3327 bp. The lengths of the proteins ranged from 755 to 1108 amino acids, and they possessed 86.96 kDa to 126.22 kDa molecular masses and predicted pI values of 5.40 to 8.52.

Combined with the classification of PLD gene families identified in *Arabidopsis thaliana* and rice, the PLD genes in three plants are divided into six subfamilies: α, β/γ, δ, ζ, φ, κ. The subfamilies φ and κ only exist in rice, while PLD genes in potato and Arabidopsis are distributed in the other four families, and 16 PLD genes in potato are distributed in α, β, δ, ζ subfamilies. The number of genes was 6, 3, 5, and 2, respectively. The member domain of α, β, and δ family was C2, and the member domain of subfamily ζ was PH-PX (Figure 1). The genetic relationship of PLD ζ was far from that of other groups, and the gene family distance of PLD φ was farther than that of other groups. The PLD family genes of potato, Arabidopsis and rice are mainly distributed in α, β and δ subfamilies, indicating that the three genes have high homology and may be similar in function. In addition, the gene distribution of potato PLD family was uneven, mainly in α and δ subfamilies, indicating that potato PLD gene family may have carried out genome replication in the process of evolution, so that the number and structure of genes among subgroups were different. By comparing the PLD protein sequence of potato and Arabidopsis, it is found that each gene contains HxKxxxxD conservative motifs at both ends of the sequence, which can indicate that the PLD protein is highly conserved among different species (Figure 2).

### 3.2. Chromosomal Location and Gene Duplication of PLD Gene Family in Potato

According to the chromosome mapping, 16 potato PLD genes were randomly and unevenly distributed on chromosomes 1, 2, 3, 4, 6, 8, 10, and 12. There was only one potato PLD gene on chromosomes 4, 6, and 12, and four PLD genes on chromosome 1, which was the chromosome with the most PLD genes, followed by chromosome 8 with three PLD genes and two PLD genes on chromosomes 2, 3, and 10 (Figure 3). There are many mechanisms of gene family amplification, including polyploidy, fragment replication, tandem replication, transposable elements, and so on [33]. To study the genomic replication event of the potato PLD gene, according to the defined standard, two tandem repeat genes were found on chromosome 8 (StPLDα4/α5).

### 3.3. Gene Structure and Cis-Acting Elements Analysis and Conserved Motif Identification

In order to further study the conservation of potato PLD protein sequence and the difference in motif composition between potato proteins, MEME was used to further analyze the conservative motif of potato PLD protein sequence. Five conserved motifs of potato PLD were identified, which were between 34 and 50 amino acids in length (Figure 4A, Table 2). The results showed that the motifs of PLDα, PLDβ, and PLDδ subfamily genes were completely consistent, and they all contain five conserved motifs. The motif composition of PLDζ subfamily genes was also completely the same, all of which contained three conserved motifs. According to the results of Pfam website, Motif1 belongs to PLDc domain, which can also be called HKD domain (PF00614), Motif5 belongs to PLD_C (PF12357.8) domain, and other conserved motifs are predicted, such as Motif3 belongs to Pilin_GH domain (PF16734). All StPLD contained the HKD domain, C2-PLD had five motifs, PXPH-PLD had three motifs, in addition to StPLDζ1 and StPLDζ2, all contained Motif 2, Motif5; the deletion of two motifs of the ζ subfamily protein makes the domain of the ζ subfamily different from the other three subfamilies. Motif1, Motif3, and Motif4 were all distributed in the four subfamilies, which indicates that the StPLD protein sequence is conserved.

According to the structure of StPLD gene, the number of exons in the StPLD gene family members ranged from 3 to 20, and all genes had introns (Figure 4B). In addition, it was also found that the gene structures of the same subfamily were similar. There were 10 exons and 9 introns in the subfamily PLDδ1~δ5, and 20 exons and 19 introns in both PLDζ1 and PLDζ2. The gene structure was completely the same. The number of exons in each gene in the same subfamily was basically the same, and there were great differences in the number of exons and introns among different subfamilies, indicating that the structure of StPLD gene is more complex.

In order to further study the regulation mechanism of StPLD gene under abiotic stress, according to the different functions of different cis-acting elements, 13 cis-acting elements related to growth and development, hormones and stress response were screened (Figure 5). The promoter of StPLD gene contained many homeopathic regulatory elements, the most StPLDα3 had 16 cis-acting elements, and the least StPLDδ1 had three cis-acting elements. There are four stress response elements, namely drought-responsive (MBS), low temperature (LTR), anaerobic induction (ARE), and light response (G-box) elements, and five hormone response elements, namely abscisic acid (ABRE), methyl jasmonate (CGTCA-motif, TGACG-motif), salicylic acid (TCA-element), auxin (TGA-element), gibberellin (TATC-box, P-box). Two growth and development response elements were meristem expression regulation (CAT-box) and circadian rhythm regulation (circadian) (Figure 5). Among the StPLD genes, G-box, ABRE, CGTCA-motif, and these three acting elements were the most, 28, 30, and 27, respectively, which were randomly distributed in each gene, indicating that these genes can participate in plant light response and respond to hormones, such as abscisic acid and methyl jasmonate. Cis-acting element analysis showed that StPLD gene was closely related to abiotic stress, growth and development and hormones in plants [34].

### 3.4. Gene Ontology Annotations of StPLD Proteins

In order to further study the biological process of StPLD protein, GO annotation was carried out by Blast2GO software, and the role of genes in the biological process (BP), molecular function (MF), and cellular component (CC) was analyzed. Through GO annotation analysis, it was found that all 16 StPLD proteins participated in the process of phospholipid catabolism, and all had phospholipase D activity (Figure 6).

In the biological process, 16 proteins were involved in the process of phospholipid catabolism. Except for StPLDζ1 and StPLDζ2, the other 14 proteins were involved in the metabolism of phosphatidylcholine. In addition, only the two proteins, StPLDζ1 and StPLDζ2, were involved in phosphatidic acid biosynthesis and inositol lipid-mediated signal transduction, and only the StPLDα6 protein was involved in water stress, salt stress, abscisic acid stress, and membrane lipid catabolism. Only the StPLDβ2 protein was involved in transcriptional regulation using DNA as a template. In molecular function, 16 StPLD proteins have phospholipase D activity, except StPLDζ1 and StPLDζ2, the other 14 proteins had a calcium ion binding function, which were distributed in StPLDα, StPLDβ, and StPLDδ, indicating that the C2 domain can better bind to cofactor Ca^2+^ and enhance the affinity of the enzyme to its substrate. Among the cellular components, 16 StPLD proteins were located in the plasma membrane, and only StPLDβ2 protein was located in the nucleus.

### 3.5. Collinear Analysis of PLD Gene in Potato and Arabidopsis thaliana

From the collinearity map of potato PLD and Arabidopsis PLD genes, it is known that there are 14 homologous gene pairs, including 14 potato PLD genes and 14 Arabidopsis PLD genes, which are distributed in four subfamilies: α, β, δ and ζ (Figure 7) indicating that there is a close homologous evolution relationship between potato and Arabidopsis PLD gene family. Among them, StPLDα1, StPLDα3, and StPLDζ2 are collinear with at least two AtPLD genes, indicating that these genes may have similar functions and play an important role in the evolution of potato and Arabidopsis PLD gene family. We found that there was no collinear relationship between StPLDα5, StPLDα6, StPLDβ1, StPLDβ2, StPLDδ1, StPLDδ5, and the AtPLD gene, indicating that these genes may be specific to potato evolution. Among these genes, there was only one single gene pair, StPLDβ2 and AtPLDγ2, indicating that this pair of genes have a common origin before potato and *Arabidopsis thaliana* gene differentiation.

### 3.6. Tissue Expression and Stress Treatment Expression Analysis of the Potato PLD Genes

In order to study the function of potato PLD gene in the process of growth and development, we obtained transcriptome data from potato genome database, found out the RNA-seq data of PLD gene in different tissues and under various stresses and drew heat maps (Table A3 and Table A4). The expression level of StPLD gene in leaves, roots, shoots, callus, stolons, tubers, flowers, petioles, petals, stamens, carpels, and sepals (Figure 8A) and the expression levels of salt, mannitol, heat, Phytophthora infestans, β-aminobutyric acid, benzothiadiazole, abscisic acid, auxin, gibberellin, and benzo (a) pyrene (Figure 8B) were revealed.

The results showed that StPLDα2, StPLDα6, and StPLDζ2 were not detected in all tissues, and StPLDα4, StPLDβ1, StPLDδ3, and StPLDζ1 were expressed in all tissues. Among them, StPLDα5 was expressed in stamens at a higher level, showing tissue-specific expression, StPLDα4, StPLDα5, StPLDβ1, StPLDδ3, StPLDζ1 were mainly expressed in roots and leaves, indicating that these genes may play a specific regulatory role in the development of roots and leaves. The expression of StPLDδ3 was significantly increased in petals, carpels, and sepals, and the expression of StPLDδ4 was significantly increased in stamens. Only StPLDα5 was not detected under P. infestans stress, while the other genes were upregulated under these 10 stresses, indicating that PLD gene family plays an important role in the regulation of potato growth and development. The gene expression of StPLDα5 increased significantly under NaCl, mannitol and 35 °C heat stress, indicating that StPLDα5 was involved in salt tolerance and drought resistance of potato, and was related to the heat tolerance mechanism of potato. StPLDα5 increased significantly under ABA hormone treatment.

### 3.7. Expression Analysis of StPLD Genes in Different Treatments

Based on the above bioinformatics analysis, the response of the potato PLD gene to biotic stress and abiotic stress was further studied. Potato Desiree materials were treated with NaCl, high temperature, drought, and ABA, and then the expression of StPLD gene was detected by real-time quantitative qRT-PCR to analyze the expression of 16 StPLD genes under NaCl, high temperature, drought, and ABA stress (Figure 9). The results showed that the expression levels of 16 genes changed under the four treatments. Among them, the expression levels of the six genes, StPLDα1, StPLDα2, StPLDδ1, StPLDδ4, StPLDδ5, and StPLDζ2, were upregulated under NaCl, high temperature, drought, and ABA stress. The expression levels of the remaining genes were both upregulated and downregulated under these four treatments.

Under NaCl treatment, StPLDα1, StPLDα4, StPLDα5, StPLDδ1, StPLDδ2, StPLDδ3, StPLDδ4, StPLDδ5, StPLDζ2 were significantly upregulated, StPLDβ2, StPLDβ3 were downregulated, and the remaining genes were upregulated but not significantly (Figure 9A), Under high temperature treatment, StPLDα1, StPLDα2, StPLDα4, StPLDα6, StPLDβ3, StPLDδ1, StPLDδ2, StPLDδ4, StPLDδ5, StPLDζ2 were significantly upregulated, and StPLDζ1 was significantly downregulated. (Figure 9B), Under drought stress, StPLDα1, StPLDα2, StPLDα5, StPLDδ1, StPLDδ3, StPLDδ4, StPLDδ5, StPLDζ2 were significantly upregulated, and StPLDζ1 was significantly downregulated. (Figure 9C). Under ABA treatment, StPLDα1, StPLDα2, StPLDβ3, StPLDδ1, StPLDδ4, StPLDδ5, StPLDζ2 were significantly upregulated, and StPLDα5, StPLDβ1, StPLDβ2, StPLDδ3, StPLDζ1 were significantly downregulated.(Figure 9D).

Under NaCl treatment, the qRT-PCR results in the PLDα subfamily, StPLDδ subfamily, and StPLDζ subfamily are consistent with the results in the RNA-seq data, and the gene expression levels were all upregulated. The qRT-PCR results of β1 and β2 in the PLDβ subfamily were consistent with the results in the RNA-seq data. The gene expression of β1 increased, and the gene expression of β2 decreased. The qRT-PCR result of β3 was different from the result in the RNA-seq data, the qRT-PCR result was decreased, and the RNA-seq data were increased.

At 38 °C, the qRT-PCR results in the PLDβ subfamily were consistent with the results in the RNA-seq data. The gene expression of β1 and β3 increased, and the expression of the β2 gene decreased. In the PLDα subfamily, the qRT-PCR results of α1, α2, α4, and α5 were consistent with the results in the RNA-seq data, and the gene expression levels were all upregulated. The qRT-PCR results of α3 and α4 were different from the results in the RNA-seq data. The qRT-PCR results of α3 and α4 increased, but the RNA-seq data decreased. In the StPLDδ subfamily, the qRT-PCR results of δ1, δ2, δ4, and δ5 were consistent with the results in the RNA-seq data, and the gene expression levels were all upregulated. The qRT-PCR result of δ3 was different from the result in the RNA-seq data, the qRT-PCR result of δ3 decreased, and the result of the RNA-seq data increased. In the StPLDζ subfamily, the qRT-PCR results of ζ2 were consistent with the results in the RNA-seq data, and the gene expression increased. The qRT-PCR result of ζ1 was different from the result in the RNA-seq data. The qRT-PCR result of ζ1 decreased, and the result of RNA-seq increased.

Under drought treatment, the qRT-PCR results in the PLDβ subfamily were consistent with the results in the RNA-seq data. The gene expression of β1 and β3 increased, and the gene expression of β2 decreased. In the PLDα subfamily, the qRT-PCR results of α1, α2, α5, and α6 were consistent with the results in the RNA-seq data, and the gene expression levels were all upregulated. The qRT-PCR results of α3 and α4 were different from the results in RNA-seq data. The qRT-PCR results of α3 increased, the RNA-seq data decreased, the qRT-PCR result of α4 decreased, and the RNA-seq data increased. In the StPLDδ subfamily, the qRT-PCR results of δ1, δ2, δ4, and δ5 were consistent with the results in the RNA-seq data, and the gene expression levels were all upregulated. The qRT-PCR result of δ3 was different from the result in RNA-seq data. The qRT-PCR result of δ3 increased, but the result of RNA-seq decreased. In the StPLDζ subfamily, the qRT-PCR results of ζ2 were consistent with the results in the RNA-seq data, and the gene expression increased. The qRT-PCR result of ζ1 was different from the result in the RNA-seq data. The qRT-PCR result of ζ1 decreased, and the result of the RNA-seq data increased.

Under ABA treatment, in the PLDα subfamily, the qRT-PCR results of α1, α2, α3, and α4 were consistent with the results in the RNA-seq data. The gene expression of α1, α2, and α4 increased, and the gene expression of α3 decreased. The qRT-PCR results of α5 and α6 were different from the results in the RNA-seq data. The qRT-PCR results of α5 and α6 decreased, and the results of the RNA-seq data increased. The qRT-PCR results in the PLDβ subfamily were consistent with the results in the RNA-seq data. The gene expression of β1 and β2 decreased, and the gene expression of β3 increased. In the StPLDδ subfamily, the qRT-PCR results of δ1, δ3, δ4, and δ5 were consistent with the results in the RNA-seq data. The qRT-PCR result of δ2 was different from the result in the RNA-seq data, the qRT-PCR result of δ2 decreased, and the result of the RNA-seq data increased. In the StPLDζ subfamily, the qRT-PCR results of ζ1 were different from the results in the RNA-seq data. The qRT-PCR results of ζ1 decreased, and the RNA-seq data increased. The qRT-PCR results of ζ2 were consistent with the results in the RNA-seq data.

## 4. Discussion

The PLD gene family has been identified in many species, including 12 in *Arabidopsis thaliana* [22], 17 in *Oryza sativa* [12], 19 in *Gossypium* [35], 13 in *Zea mays* [3], and 13 in *Cicer arietinum* [14]. This experiment identified 16 PLD genes in the potato genome, all of which contained HxKxxxxD conserved motifs, with molecular weights ranging from 86.96 kDa to 126.22 kDa (Table 1). Compared with the above-mentioned species, the potato PLD gene has not expanded significantly, only two genes were tandem repeat genes. These 16 PLD gene family members were randomly distributed on potato chromosomes 1, 2, 3, 4, 6, 8, 10, and 12 (Figure 3). Based on the phylogenetic analysis of *Arabidopsis thaliana* and rice, it was found that 16 potato PLD were divided into four subfamilies: α, β, δ, and ζ (Figure 1). PLD α was the most common in potato PLD gene family, which was similar to the subfamily classification of PLD gene in *Arabidopsis thaliana*, rice, and maize.

The difference is that the phylogenetic analysis of the PLD gene family in rice shows that there are two more subfamilies than potato and *Arabidopsis thaliana*, namely κ and φ, and two more subfamilies in cotton, φ and ε (Figure 1), which is caused by the different N-terminal domains of PLD. According to the different domains, it can be divided into three subfamilies: C2-PLD, PX/PH-PLD, and SP-PLD. α, β, δ, γ, and ε were classified into the C2-PLD subfamily, PLDζ was classified into the PX/PH-PLD subfamily, and PLDφ was classified into the SP-PLD subfamily (Figure 4) [12,13,36]. Collinear analysis showed that there were 14 pairs of homologous genes between potato PLD and Arabidopsis PLD. Except for StPLDβ2 and AtPLDγ2, each pair of collinear genes were in the same subfamily (Figure 7), indicating the conservatism of genes in the same group of StPLD. At present, there are few reports on the collinear genes of PLD gene family, and their functions need to be further studied.

The difference in gene structure plays an important role in the evolution of gene families. Based on the analysis of the gene structure of potato and *Arabidopsis thaliana* PLD gene, it was found that the gene members of the same subfamily had similar exon and intron structures (Figure 4B), but they were not exactly the same, indicating that functional differentiation had taken place in potato PLD gene in the long process of evolution [37]. Conservative motif analysis showed that the number and sequence of motifs in the same subfamily were similar; only the motifs in the PLD ζ subfamily were different, which may be due to the acquisition or loss of conserved motifs in the PLD gene family in the process of evolution [38]. The C2-PLD subfamily had five motifs, and the PXPH-PLD subfamily had three motifs. Motif 2 contained a regular-expression sequence “IYIENQ[F/Y]F”. The seventh amino acid of this sequence, Phenylalanine (F), appeared in all PX/PH-PLDs, but was often substituted by Tyrosine (Y) in the C2-PLDs [39]. Motif 4 contained a highly conserved core triad “ERF” in the C2-PLDs [40].

The promoter area containing the cis-action element is considered to be related to gene expression of plant growth and adaptation [41]. It is predicted that its cis-acting element is displayed, and the potato PLD promoter region contains many acts associated with non-biological stress and hormone response, such as drought (MBS), low temperature (LTR), light response (G-Box), abscisic acid (ABRE), Salicylic acid (TCA-element) (Figure 5), etc. It showed that these genes are regulated by low temperature and drought and also respond to the influence of exogenous hormones on plants, validated by qRT-PCR. qRT-PCR gene expression was basically in line with the prediction of cis-acting elements. StPLD gene expression changed under drought and ABA treatment. Among all the cis-acting elements, ABRE had the largest number of elements, indicating that most of the potato PLD members may be related to the stress regulated by abscisic acid. Abscisic acid responds to abiotic stress and also regulates stomata closure and root growth [42]. The subsequent real-time quantitative results showed that compared with the control, the gene expression of the tissue-cultured plantlets treated with abscisic acid was upregulated. It is inferred that ABA plays an important role in the growth and development of potatoes.

Detecting the expression of StPLD gene by real-time quantitative qRT-PCR found that when plants were exposed to drought, high temperature, salt stress, and other adversities, they used a variety of methods to deal with adversities. At this time, the mechanism of plant genes shows similarities [43]. In this experiment, the expression patterns of 16 StPLD genes under three abiotic stress and one hormone treatment were analyzed. The expression levels of all genes under the four treatments showed different degrees of increase and decrease (Figure 9), which were consistent with the function of the cis-acting element predicted above. The quantitative results showed that the expression of 11 genes increased the most under high temperature treatment, which was similar to the previous study that BrPLD of cabbage [44] had upregulated expression of nine genes under high temperature treatment at 37 °C. One CbPLD was cloned in Alpine ion mustard. A CbPLD gene was cloned in Chorispora bungeana, and it was found that CbPLD was induced at 40 °C, indicating that the gene may be regulated by high temperature [45]. In Jatropha curcas, the enzyme activity of JcPLDα increases with temperature and reaches a peak at 60 °C, which also indicates that the JcPLDα gene is affected by high temperature [46]. These results all indicate that the PLD gene family can regulate plant growth and development when plants are under high temperature stress. In addition, under low temperature stress, PLD may be involved in the disintegration of plasma membrane and the change in membrane fluidity. AtPLDδ has been proven to be a positive regulator of freezing stress tolerance [47].

Studies have shown that under 20% polyethylene glycol (PEG6000) simulated drought stress, the expression of four genes in the aboveground part is significantly upregulated, and the expression of nine genes in the underground part is significantly upregulated. Under ABA treatment, the expression levels of seven genes were significantly upregulated in the aboveground part, and the expression levels of six genes in the underground part were significantly upregulated [3]. The quantitative results of StPLDs showed that the expression levels of 11 genes were significantly upregulated under drought stress (Figure 9C), and the expression levels of seven genes were significantly upregulated under ABA treatment (Figure 9D), which was similar to the quantitative results of maize. Under drought stress, plants can adapt by regulating stomata. GPA1 protein controls the activity of PLDα, and GPA1 protein and PLDα1 influence the process of ABA-induced stomata opening and closing through interaction [39]; it shows that PLDα1 plays an important role in regulating the opening and closing of stomata. The quantitative results under drought stress showed that not only the genes of the PLDα subfamily were significantly upregulated, but the genes of other sub-families were also significantly upregulated, indicating that other sub-families are also involved in the process of stomata opening and closing [47]. Studies have shown that chickpea PLD [14] genes are exclusively upregulated in the upper part of the ground under drought stress, indicating that they may play a role in the closure of stomata, thereby reducing transpiration and water loss. Prior to this, PLD has been thought to regulate stomata closure under drought and hypertonic stress [48,49]. These are enough to show that the PLD gene family plays an important role in regulating the opening and closing of plant stomata in response to drought stress [50]. In *Arabidopsis thaliana* under salt stress, PLDα1 and its derivative PA regulate the mitogen-activated protein kinase (MAPK) signal components MAPK6 and MAPK3 [51]. MAPK6 is activated by PLDα1-derived PA and interacts with SOS1(Na^+^/H^+^ reverse transport protein) and phosphorylates it. Phosphorylation leads to the activation of SOS1. The activated SOS1 effectively eliminates excess Na+ from the cell, thereby preventing the cell’s Na^+^ toxicity [52]. In addition, PLDα-1 derived PA also binds to the microtubule-associated protein MAP65-1, which helps stabilize the microtubules, thereby improving salt tolerance [53].

The tissue expression pattern of genes is closely related to functional characteristics [31]. This study used potato tissue expression transcript data to analyze the expression of the StPLD family in 12 tissues. Potato PLD genes were abundantly expressed in petals, stamens, carpels, and sepals (Figure 8A), indicating that the expression level of PLD genes in the flowering period is higher than other periods, in particular, the three genes StPLDα5, StPLDβ1, and StPLDδ4 were significantly expressed in the stamens. It can be speculated that these three genes may prolong or advance the flowering period of potato. Using qRT-PCR technology to detect the gene expression changes of potato PLD gene under drought, high temperature, salt stress, and ABA treatment, expression profile analysis showed that the expression of StPLDβ2 and StPLDβ3 decreased under salt stress, and the expression of StPLDα5 decreased under drought stress, which was consistent with the subsequent quantitative results.

## 5. Conclusions

To sum up, this study identified 16 StPLD genes from the potato genome, and all had HxKxxxxD domains. They are divided into four subfamilies: α, β, δ, and ζ according to the phylogenetic relationship. StPLDs in the same subfamily had similar gene structures and conserved motifs. Two tandem repeat genes were found in the PLD gene family. According to different structural domains, the StPLD gene family can be divided into two main subfamilies, C2-PLD and PXPH-PLD. Potato PLD genes were distributed on eight potato chromosomes. The expression levels of StPLD genes under salt, high temperature, drought, and ABA treatments were analyzed by qRT-PCR, which provided references for subsequent functional verification of these genes.

## Figures and Tables

**Figure 1 biology-10-00741-f001:**
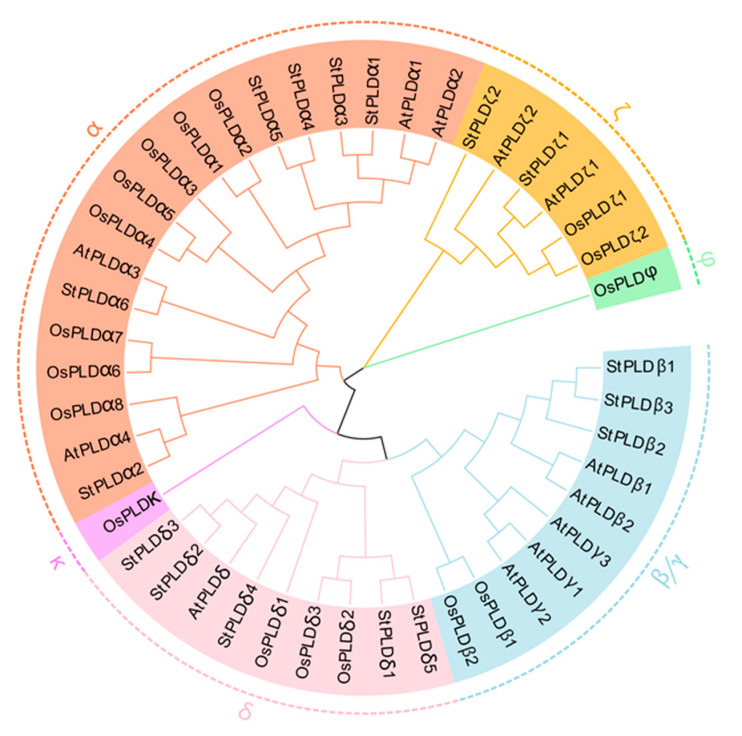
Phylogenetic analyses of the plant PLD proteins. The conserved PLD proteins from potato, rice, Arabidopsis were aligned using ClustalW2.0, and the phylogenic tree was constructed using the NJ method with bootstrapping analysis (1000 replicates).

**Figure 2 biology-10-00741-f002:**
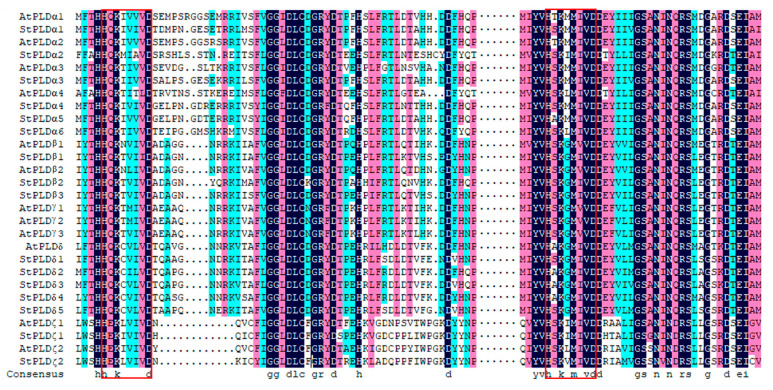
Alignment of potato and Arabidopsis family protein sequences. Amino acids marked in dark blue indicate 100% sequence identity, pink indicates ≥75% identity, and light blue indicates ≥50% identity. The red boxes indicate the HxKxxxxD domain of the PLD gene family.

**Figure 3 biology-10-00741-f003:**
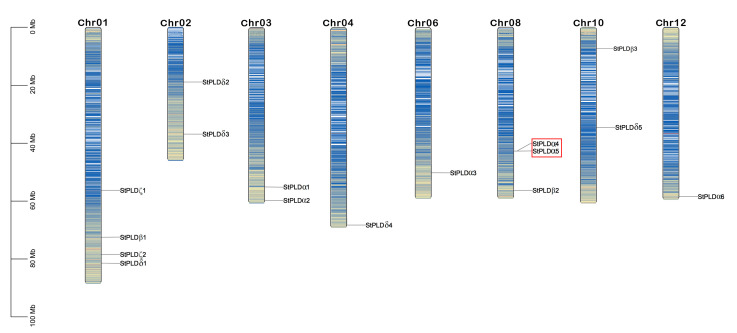
Chromosomal distribution of Potato PLD genes. Red box indicates tandem duplicate genes. Blue lines on chromosomes indicate gene density.

**Figure 4 biology-10-00741-f004:**
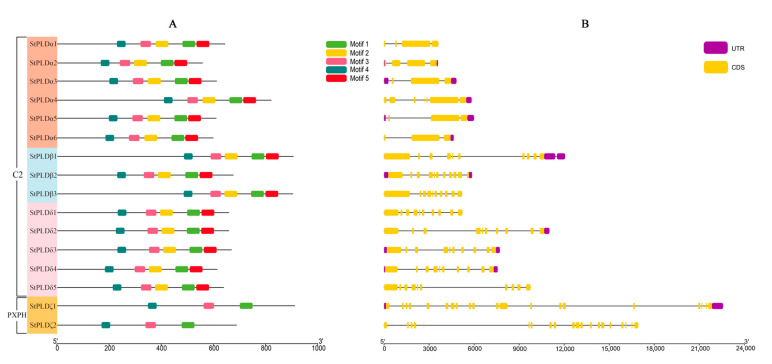
Conserved motif (**A**) and gene structure (**B**) analysis of StPLD genes. Different motifs are indicated by different colors. Yellow color bars represent the exon, lines represent the intron, while purple color bars indicate the untranslated region (UTR) both at 5′ and 3′ ends.

**Figure 5 biology-10-00741-f005:**
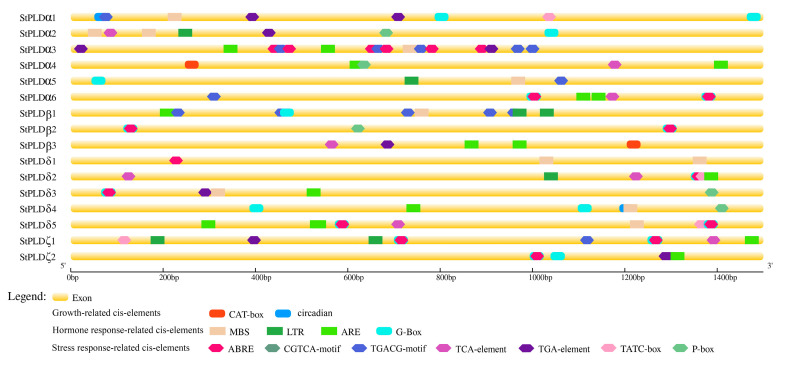
The cis-acting element of 1500 bp sequences upstream of the potato StPLD gene. This study used the database PlantCARE to predict the motif.

**Figure 6 biology-10-00741-f006:**
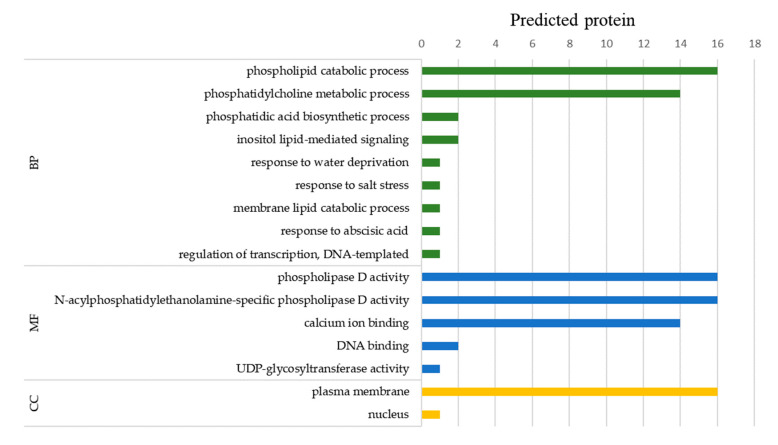
Gene ontology (GO) annotation of StPLD proteins. The gene ontology (GO) analysis of potato PLD proteins was performed using the Blast2GO software (https://www.blast2go.com/ (accessed on 26 November 2020)). The annotation was performed on three categories, biological process (BP), molecular function (MF), and cellular component (CC). The numbers on the abscissa indicate the number of predicted proteins.

**Figure 7 biology-10-00741-f007:**
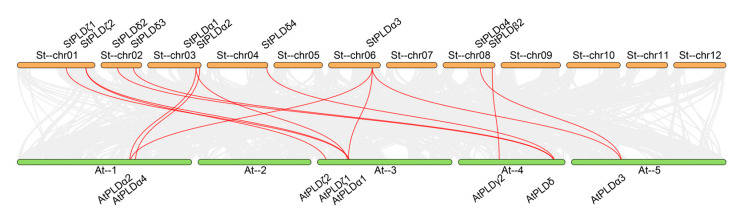
Homology analysis of PLD genes between potato and Arabidopsis. Gray lines: Collinear blocks between potato and Arabidopsis genomes. Red lines: Syntenic PLD gene pairs between potato and Arabidopsis.

**Figure 8 biology-10-00741-f008:**
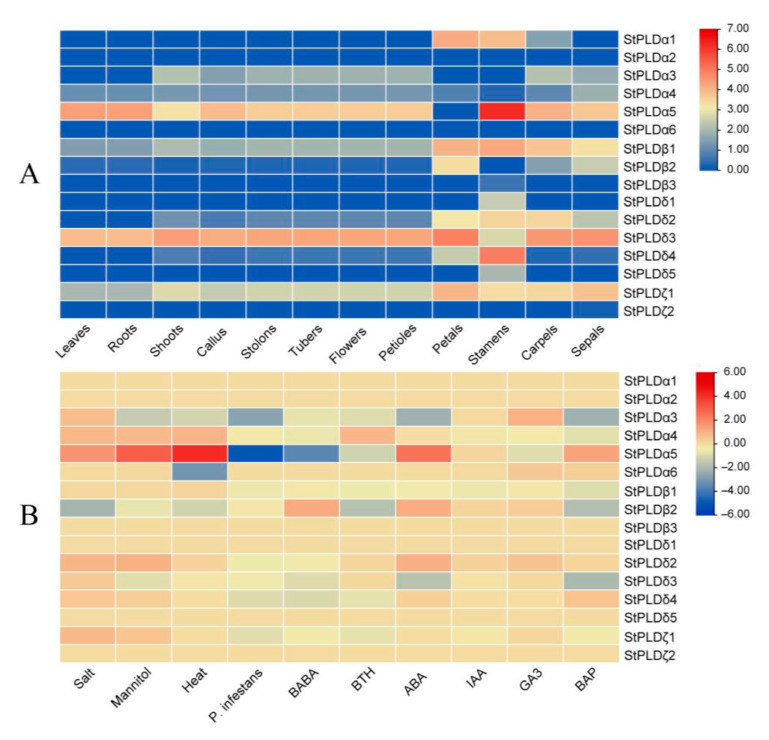
(**A**) Heatmap shows the expression of StPLD gene in 12 tissues, including the leaves, roots, shoots, callus, stolons, tubers, flowers, petioles, petals, stamens, carpels, sepals. Red indicates high relative gene expression, whereas blue indicates low relative gene expression. (**B**) Heatmap of the expression profile of potato PLD genes under 10 different biotic or abiotic stresses. Abiotic stresses included salt, mannitol, and heat; biotic stresses included DL-b-amino-n-butyric acid (BABA), stress-elicitors acibenzolar-S-methyl (BTH), and Phytophthora infestans; and other stress responses were mainly induced by four plant hormones: 6-benzylaminopurine (BAP), indole-3-acetic acid (IAA), abscisic acid (ABA), and gibberellic acid (GA3). Red indicates gene upregulation, while blue indicates gene downregulation.

**Figure 9 biology-10-00741-f009:**
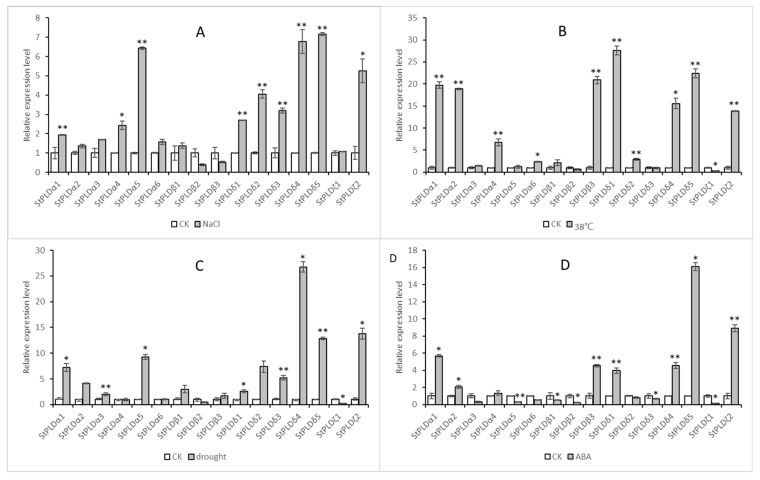
Differential expression of the StPLD gene family in response to abiotic stress and hormone induction. (**A**) Relative expression level under NaCl treatment. (**B**) Relative expression level at 38 °C. (**C**) Relative expression level under drought treatment. (**D**) Relative expression level under ABA treatment. (* t-test *p*-value < 0.05, ** t-test *p*-value < 0.01).

**Table 1 biology-10-00741-t001:** The PLD genes in potato and properties of the deduced proteins (*Solanum tuberosum*).

Gene ^1^	Gene ID ^1^	Chromosome Location (bp) ^1^	ORF Length (bp) ^1^	No. of Extrons	Protein ^2^			Gene Subfamily
					Length (aa)	MW (Da)	pI	
StPLDα1	Soltu.DM.03G030690	chr03:55077171..55080788(+)	2526 bp	4	841	96,517.87	5.40	C2
StPLDα2	Soltu.DM.03G036240	chr03:59268234...59271821(+)	2268 bp	4	755	86,957.41	8.45	C2
StPLDα3	Soltu.DM.06G023780	chr06:50043190...50048016(+)	2430 bp	3	809	92,232.31	5.42	C2
StPLDα4	Soltu.DM.08G015290	chr08:42699245...42705076(+)	3057 bp	7	1018	11,6148.05	6.14	C2
StPLDα5	Soltu.DM.08G015300	chr08:42713576...42719569(+)	2424 bp	3	807	91,963.58	5.71	C2
StPLDα6	Soltu.DM.12G028870	chr12:58460700...58465340(+)	2391 bp	3	796	90,670.03	6.00	C2
StPLDβ1	Soltu.DM.01G032650	chr01:72447542...72435483(−)	3312 bp	10	1103	122,968.20	6.62	C2
StPLDβ2	Soltu.DM.08G026790	chr08:56273785...56279647(+)	2622 bp	11	873	98,464.98	7.66	C2
StPLDβ3	Soltu.DM.10G006880	chr10:7384934...7390130(+)	3306 bp	10	1101	123,360.69	6.72	C2
StPLDδ1	Soltu.DM.01G042810	chr01:81140965...81135745(−)	2571 bp	10	856	95,751.91	6.27	C2
StPLDδ2	Soltu.DM.02G005480	chr02:18854673...18843653(−)	2571 bp	10	856	98,063.45	6.39	C2
StPLDδ3	Soltu.DM.02G023290	chr02:36855596...36847884(−)	2601 bp	10	866	98,679.36	6.51	C2
StPLDδ4	Soltu.DM.04G037130	chr04:68165888...68173456(+)	2436 bp	10	811	92,287.16	7.40	C2
StPLDδ5	Soltu.DM.10G012190	chr10:34475195...34465438(−)	2511 bp	10	836	93,725.06	8.52	C2
StPLDζ1	Soltu.DM.01G020480	chr01:56213337...56190748(−)	3327 bp	20	1108	126,223.72	6.39	PH-PX
StPLDζ2	Soltu.DM.01G039740	chr01:78401722...78418660(+)	2658 bp	20	885	101,538.59	6.00	PH-PX

^1^ Gene information was retrieved from the S. tuberosum v6.1 genome annotation (http://solanaceae.plantbiology.msu.edu/dm_v6_1_download.shtml (accessed on 21 November 2020)). ^2^ Protein profiles information from the ExPASy-ProtParam tool (https://web.expasy.org/protparam/ (accessed on 21 November 2020)).

**Table 2 biology-10-00741-t002:** List of the conserved motifs of StPLD proteins.

Motif	Length	Amino Acid Sequence
Motif1	23	KFRRFMIYVHSKGMIVDDEYVIIGSANINQRSLDGSRDTEIAMGAYQPHH
Motif2	29	MEJALKIASKIRAGERFAVYIVVPMWPEGLPESASVQEILFWQRRTMQMM
Motif3	33	VSGKNLIIDRSIHDAYIKAIRRAQHFIYIENQYFJGSSYSW
Motif4	21	PREPWHDJHCRIEGPAAYDVLYNFEQRWRKAGKW
Motif5	29	QEPPRGQIYGYRMSLWAEHLGMLEDCFQHPESLECVRRVNEIAEKNWKQY

## Data Availability

Not applicable.

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
