# Peer review of "Genome-Wide Analysis and Expression Profiling of the Phospholipase D Gene Family in Solanum tuberosum"

_biology, 2021, doi:10.3390/biology10080741_

Round 1

Reviewer 1 Report

Simple Summary: The importance and function of the phospholipase D gene family should be explained first and then the summary of your analysis should take place.

Introduction:

  • Arabidopisis thaliana should be in italics throughout the manuscript unless you use only the word Arabidopsis.
  • Line 60: dicotyledons should be capitlised.
  • Line 87: Was it highly expressed in pollen tubes or not? Confusing.

Materials & Methods:

  • Why were the seedlings grown in liquid medium? Explain.
  • Line 103: Change “then treated differently.” To something like: “in preparation for the different experimental treatments.”
  • Line 115: Break up the hyperlink.
  • Line 128: remove the hyphen in “neighbor joining”.
  • Line 133 to 153: This whole sentence needs rephrasing, hard to understand.
  • Line 149: State that this analysis was done.
  • Line 150: Change the tense of “Use” to past tense.
  • The list of primer sequences should be mentioned in the qPCR section.

Results:

  • Neaten up Table 1, especially the brackets that aren’t aligned.
  • Figure 2: What do the red boxes indicate?
  • Figure 3: Full stop after genes (line 217).
  • Figure 6: Heading should say “predicted” protein. Also, what do the abbreviations on the side mean? Spell out in figure legend.
  • NACL should be NaCl.
  • Some explanation of the experimental conditions of the RNAseq data is required. Were the different tissues sampled at the same time and snap frozen? What was the growth stage the samples were collected at?
  • Figure 8: Should have a citation of the database where you got that data.
  • Line 338: Should be “biotic” stress.
  • Figure 9: Should the Chinese writing be present? What is the reference gene used for the RT-PCR? Are any of these upregulated genes significantly upregulated? If so, that should be highlighted on the figure.
  • Does the qPCR results mirror those found in the RNAseq database? Should compare and contrast in the results section.

Discussion:

  • Line 366: rephrase sentence.
  • Line 371: How do you know their chromosomal distribution is random?
  • Line 407: Playing the function of against adversity?
  • Line 409 to 412: re-write or rephrase sentence.
  • Line 438: ground?
  • Line 443: When you state that the quantitative results are linked to stomata, you don’t justify your sentence. Only later in the paragraph do you cite research that backs up your claim.  Re-structure this section so it is more obvious to the reader what you are trying to say.

Reviewer 2 Report

The manuscript describes the study carried out by the authors to identify 16 phospholipase D genes in potato (Solanum tuberosum). Authors show an expression analysis of their transcripts in different tissues and under salt, high temperature, drought and ABA treatments. The manuscript needs considerable improvement for publication.

Materials and methods must describe in detail how the tissue expression analysis was performed and under the studied treatments. Which controls were used? Which gene was used to normalize the results? How many replicates were used? Which tissue was used for the expression analysis under the stress conditions?

Results and discussion need a better writing and explanation of the results obtained based on the scientific evidence. Some examples not being exhaustive:

-Figure 2 and 5 not accurately described and not mention in the text.

-Some sentences are not understandable. E.g: Lines 224-225; Lines 436-439.

-Language is imprecise. E.g: Line 213 and so on.

Reviewer 3 Report

The manuscript addresses an issue that is very important about the potato phospholipase D gene family. However, there are some important issues that the authors need to address before the manuscript can be considered for publication. The following are my comments describing these issues.

Line 37. Please, "phospholipase D” keyword, must be changed because they are found in the title.

The Introduction has to be restructured. Introduction should be reasonably brief and its paragraphs should be placed in a logical sequence so that together they form a coherent unit without any repetition.

Line. 57. The authors say: At present .... but the references are from the years 2002, 2007, and 2010? In Addition, use scientific names for each species.

Line 92. The authors investigated with solanum tuberosum seedlings. Is there any special reason for this choice? because in the introduction this information is unclear. What is their main hypothesis? Authors mention some crops such as Oryza sativa, Arabidopsis thaliana, Vitis vinifera and other, but there is no background on solanum tuberosum, why?

Line 98. Please, include the location (coordinates) and when the study was conducted.

Line 100-103. Which solid medium? and which liquid medium (nutrient solution)? Please, provide more details here.

Line 108. Why this temperature and not another? Justify the treatments used

Untreated plantlets were used as controls?

How many replicates for each condition?

How many potato plantlets were used for each condition?

In general methods need more detail. This lack of detail makes it difficult to understand what was done and thus to interpret/understand your results.

Round 2

Reviewer 2 Report

The reviewer appreciates the effort the authors did to improve the manuscript. I recommend the publication of the current version of the manuscript. Two minor comments:
-In tables A3/A4: the title says ' FPKM values of auxin transporter genes in various tissues'. I guess this is a mistake.
-Figure 9: the current version is not as sharp as in the original version of the manuscript. Also please indicate what it is CK as referring to white bars.